# Prognostic Significance of Aberrant Claudin-6 Expression in Endometrial Cancer

**DOI:** 10.3390/cancers12102748

**Published:** 2020-09-24

**Authors:** Manabu Kojima, Kotaro Sugimoto, Mizuko Tanaka, Yuta Endo, Hitomi Kato, Tsuyoshi Honda, Shigenori Furukawa, Hiroshi Nishiyama, Takafumi Watanabe, Shu Soeda, Keiya Fujimori, Hideki Chiba

**Affiliations:** 1Department of Basic Pathology, Fukushima Medical University School of Medicine, Fukushima 960-1295, Japan; m2149@fmu.ac.jp (M.K.); mizuko@fmu.ac.jp (M.T.); hiace_tokotsu@yahoo.co.jp (Y.E.); m151037@fmu.ac.jp (H.K.); 2Department of Obstetrics and Gynecology, Fukushima Medical University School of Medicine, Fukushima 960-1295, Japan; s-furu@infoseek.jp (S.F.); wata@fmu.ac.jp (T.W.); s-soeda@fmu.ac.jp (S.S.); fujimori@fmu.ac.jp (K.F.); 3Department of Obstetrics and Gynecology, Iwaki City Medical Center, Iwaki 973-8402, Japan; boo.honda@gmail.com (T.H.); nishi-ya@fmu.ac.jp (H.N.)

**Keywords:** claudin, CLDN6, endometrial cancer, prognosis, tight junction, biomarker

## Abstract

**Simple Summary:**

The claudin (CLDN) family, the backbone of tight junctions, consists of more than 20 members in humans, and exhibits distinct expression patterns in tissue- and cell-type-specific manners. Among the CLDN members, CLDN6 is primarily expressed in diverse embryonic epithelial cells. It is also aberrantly expressed in various types of cancers, but its significance remains obscure. In the present study, we generated a highly specific anti-human CLDN6 monoclonal antibody, and assessed the prognostic significance of aberrant CLDN6 expression in endometrial cancer tissues. This study indicates that high CLDN6 expression in endometrial cancer relates to several clinicopathological factors and is an independent prognostic factor. The established monoclonal antibody could be a valuable tool to evaluate CLDN6-expressing tumors.

**Abstract:**

Background: Among the claudin (CLDN) family, CLDN6 exhibits aberrant expression in various cancers, but its biological relevance has not yet been established. We generated a monoclonal antibody (mAb) against human CLDN6 and verified its specificity. By immunohistochemical staining and semi-quantification, we evaluated the relationship between CLDN6 expression and clinicopathological parameters in tissues from 173 cases of endometrial cancer. Results: The established mAb selectively recognized CLDN6 protein. Ten of the 173 cases (5.8%) showed high CLDN6 expression (score 3+), whereas 19 (11.0%), 18 (10.4%) and 126 (72.4%) cases revealed low CLDN6 expression (score 2+, 1+ and 0, respectively). In addition, intratumor heterogeneity of CLDN6 expression was observed even in the cases with high CLDN6 expression. The 5-year survival rates in the high and low CLDN6 groups was approximately 30% and 90%, respectively. Among the clinicopathological factors, the high CLDN6 expression was significantly associated with surgical stage III/IV, histological type, histological grade 3, lymphovascular space involvement, lymph node metastasis and distant metastasis. Furthermore, the high CLDN6 expression was an independent prognostic marker for overall survival of endometrial cancer patients (hazard ratio 3.50, *p* = 0.014). Conclusions: It can be concluded that aberrant CLDN6 expression is useful to predict poor outcome for endometrial cancer and might be a promising therapeutic target.

## 1. Introduction

Endometrial cancer is the most common gynecological malignancy in developed countries, with an increased prevalence worldwide [1]. Although it has been considered to occur during the postmenopausal period, cases diagnosed in premenopausal women are growing [2,3]. The risk factors for endometrial cancer include an excess of endogenous and exogenous estrogens, older age, obesity and nulliparity [4,5,6]. Patients with endometrial cancer are often found at the early stages and have a relatively favorable prognosis. However, up to 20% of cases recur after primary surgery, and the 5-year overall survival rates for International Federation of Gynecology and Obstetrics (FIGO) stages III and IV cancer are 57–66% and 20–26%, respectively [7]. Therefore, biomarkers that reflect the malignant behavior of endometrial cancer are required in order to identify patients with poor outcome.

Claudins (CLDNs) are major proteins of tight junctions, and the apical-most components of apical junctional complexes [8,9,10,11]. The CLDN family is composed of 24 members in humans, and displays distinct expression patterns in tissue- and cell-type-selective manners. CLDNs also show aberrant expression in a variety of cancer tissues [12,13,14,15]. These tetraspanning membrane proteins have a short cytoplasmic N-terminus, two extracellular loops (EC1 and EC2) and a C-terminal cytoplasmic domain. CLDNs act as paracellular barriers or pores via the EC1 to regulate the selective transport of ions and substances. In contrast, CLDN EC2 participates not only in the binding of *Clostridium perfringens* enterotoxin (CPE), but also in *trans*-interaction between the plasma membranes of neighboring cells. Furthermore, the C-terminal cytoplasmic domain of CLDNs is thought to propagate intracellular signals, but the underlying molecular basis has not been determined [16].

Among the CLDN family, CLDN6 is expressed in several types of embryonic epithelial cells but not largely in normal adult cells [17,18,19,20,21]. We previously demonstrated that CLDN6-mediated cell‒cell adhesion induces epithelial differentiation in mouse F9 and embryonic stem cells [21]. CLDN6 is also highly expressed in germ cell tumors, including seminomas, embryonal carcinomas and yolk sac tumors, as well as in some cases of gastric adenocarcinoma, lung adenocarcinoma, ovarian adenocarcinoma and endometrial carcinoma [22,23]. However, the biological significance of CLDN6 expression in these cancers remains unclear.

In the present study, we developed a novel monoclonal antibody (mAb) that selectively recognizes CLDN6. Using this specific mAb, we show that the high CLDN6 expression in endometrial cancer is significantly associated with several clinicopathological factors. We also demonstrate that aberrant CLDN6 expression is an independent prognostic marker for endometrial cancer.

## 2. Results

### 2.1. Establishment of an Anti-Human CLDN6 mAb

We first generated a novel mAb against the C-terminal cytoplasmic region of human CLDN6 (Figure 1A) using the iliac lymph node method [24]. Among 384 hybridomas, 24 clones were selected by enzyme-linked immunosorbent assay (ELISA), 20 of which were able to detect CLDN6 by Western blotting in HEK293T cells transfected with the corresponding expression vector (Figure 1B,C). To check the specificity of an anti-human CLDN6 mAb (clone #15) and the previously established anti-mouse CLDN6 polyclonal antibody (pAb; [25]), HEK293T cells were transiently transfected with individual CLDN expression vectors, followed by Western blotting and immunohistochemical analyses. Clone #15 selectively recognized CLDN6 but not CLDN1, CLDN4, CLDN5 or CLDN9, which are closely related to CLDN6 within the CLDN family (Figure 1D,E). In contrast, the anti-CLDN6 pAb reacted not only with CLDN6 but also with overexpressed CLDN4 and CLDN5 to a lesser extent. We also clarified the complementarity-determining regions (CDRs) of clone #15 (Figure 1F).

### 2.2. Differential Expression of CLDN6 among Endometrial Cancer Subjects

Using immunohistochemistry, we next evaluated the expression of CLDN6 in endometrial cancer tissues resected from 173 patients (Table 1). Based on semi-quantification using the immunoreactive score (Table 2), 10 of the 173 cases (5.8%) showed high CLDN6 expression (score 3+). Among the low-expression group, 19 (11.0%), 18 (10.4%) and 126 (72.8%) cases had scores of 2+, 1+ and 0, respectively.

CLDN6 was primarily distributed along the cell membranes of endometrial carcinoma cells, and the signal intensity (SI) appeared to vary among endometrial cancer subjects (Figure 2A). Interestingly, CLDN6 exhibited intratumoral heterogeneity, and CLDN6-positive and negative subpopulations were observed in endometrial cancer tissues even in the subjects with high CLDN6 expression (Figure 2B).

### 2.3. High Expression of CLDN6 Correlates with Poor Prognosis and Several Clinicopathological Features in Endometrial Cancer

Kaplan–Meier plots revealed significant differences in overall survival and recurrence-free survival between the two groups (Figure 3A,B). The 5-year survival rate in the high CLDN6 expression group remained at approximately 30%, whereas that in the low expression group was 90%.

Among the clinicopathological factors, high CLDN6 expression was significantly associated with surgical stages III/IV (*p* ≤ 0.001), histological type (*p* = 0.030), histological grade 3 (*p* = 0.004), lymphovascular space involvement (LVSI; *p* = 0.005), lymph node metastasis (*p* = 0.001) and distant metastasis (*p* < 0.001) (Table 3). In contrast, aberrant CLDN6 expression did not relate to younger age (*p* = 0.122).

### 2.4. Aberrant CLDN6 Expression Is an Independent Prognostic Marker for Endometrial Cancer

We subsequently performed Cox multivariable analysis in endometrial cancer subjects to verify the independent predictors of survival. Among the analyzed variables, stages III/IV (hazard ratio (HR) 10.93, *p* = 0.002), distant metastasis (HR 4.68, *p* = 0.006) and high CLDN6 expression (HR 3.50, *p* = 0.014) were independent prognostic variables for overall survival of endometrial cancer patients (Table 4). In contrast, younger age, histological grade, LVSI or lymph node metastasis were not independent prognostic markers for endometrial cancer subjects.

## 3. Discussion

In the present study, we demonstrated that high CLDN6 expression in endometrial cancer tissues, in which the strong and moderate signal intensity (SI) on cell membranes was observed at greater than 30% and 50%, respectively, was significantly related to several clinicopathological features such as surgical stages III/IV, histological type, histological grade 3, LVSI, lymph node metastasis and distant metastasis. Importantly, high CLDN6 expression was an independent prognostic factor (HR 3.50), and the 5-year survival rate was about 30%, which was one third of that in the low-expression group. Thus, aberrant CLDN6 expression appeared to correlate with poor outcome in patients with endometrial cancer. Analysis of a larger number of cases would be required to draw more solid conclusions about the clinicopathological relevance of the high CLDN6 expression in endometrial cancer subjects. The relationship between CLDN6 expression and the Cancer Genome Atlas (TGCA) molecular classification of endometrial cancer may be also considered in future experiments. It is unknown how the high CLDN6 expression leads to poor prognosis in endometrial cancer subjects. However, we have recently uncovered that the EC2-dependent engagement of CLDN6 recruits and activates Src-family kinases, which in turn phosphorylate CLDN6 at Y196/200 and propagate the PI3K/AKT pathway, and this signaling axis stimulates the retinoic acid receptor γ (RARγ) and estrogen receptor α (ERα) activity [26]. Association between CLDN6 expression and PI3K/AKT activity in the endometrial cancer cell line HEC-1B is also reported, although the underlying molecular basis is not defined [27]. Taken together with the notion that ERα acts as a master transcription factor in endometrial cancers [28], aberrant CLDN6 signaling may promote the malignant behavior of endometrial cancer cells via hijacking the CLDN6–ERα pathway.

CLDNs comprise a gene family as described above, and some anti-CLDN Abs are known to react not only with the corresponding CLDN but also with other CLDN subtypes [29]. Therefore, it is of particular importance to verify the specificity of the anti-CLDN Abs used. Along this line, we previously established the anti-CLDN pAbs that selectively recognize CLDN1, CLDN5, CLDN6, CLDN7, CLDN12 or CLDN15, as far as we determined [25,30,31]. The anti-CLDN6 pAb is one of the most reliable anti-CLDN6 Abs, and is used for immunohistochemical staining of formalin-fixed, paraffin-embedded human tissues [23,32,33]. However, we noticed in the present work that it also reacted with highly expressed CLDN4 and CLDN5. The cross-reactivity of the anti-CLDN6 pAb is less efficient than the binding to CLDN6, yet it reinforces the importance of validating the selectivity of each anti-CLDN Ab. Considering the higher specificity of the novel anti-human CLDN6 mAb and the above-mentioned finding that CLDN6 is barely expressed in normal adult cells, the established mAbs could provide powerful tools that selectively recognize CLDN6 protein in a range of cancer tissues.

Genomic and non-genomic heterogeneity among distinct cell populations within cancers is known to influence tumor behavior [34]. Our immunohistochemical study revealed intratumoral heterogeneity of CLDN6 expression within human endometrial cancer tissues. These tumors were composed of CLDN6-positive and -negative subpopulations, even in endometrial cancer tissues with high CLDN6 expression. Hence, the expression of CLDN6 should be carefully evaluated when small biopsy specimens and tissue arrays are subjected to immunohistochemistry.

In summary, we here established an mAb that specifically reacts with CLDN6. We also demonstrated that aberrant CLDN6 expression is an independent prognostic variable for endometrial cancer. Therefore, CLDN6 may be a promising therapeutic target for endometrial cancer. Along this line, chimeric antigen receptor (CAR)-T cell therapy could be an attractive tool to treat CLDN6-expressing tumors [33]. It would also be interesting to determine the biological relevance of the high CDLN6 expression in various types of cancers.

## 4. Materials and Methods 

### 4.1. Generation of Antibodies

A rabbit pAb against CLDN6 was generated in cooperation with Immuno-Biological Laboratories (#18865) as described previously [25].

Rat mAbs against CLDN6 were established using the iliac lymph node method [24]. In brief, a polypeptide, (C)SRGPSEYPTKNYV, corresponding to the cytoplasmic domain of CLDN6, was coupled via the cysteine to Imject^TM^ Maleimide-Activated mcKLH (Thermo Fisher Scientific, Waltham, MA, USA). The conjugated peptide was intracutaneously injected with Imject^TM^ Freund’s Complete Adjuvant (Thermo Fisher Scientific) into the footpads of anesthetized eight-week-old female rats. All animal experiments complied with the National Institutes of Health Guide for the Care and Use of Laboratory Animals, and were approved by the Animal Committee at Fukushima Medical University (approval code: 29098). The animals were sacrificed 14 days after immunization, and the median iliac lymph nodes were collected, followed by extraction of lymphocytes by mincing. Extracted lymphocytes were fused with cells of the SP2 mouse myeloma cell line by polyethylene glycol. Hybridoma clones were maintained in GIT medium (Wako, Osaka, Japan) with supplementation of 10% BM-Condimed (Sigma–Aldrich, St. Louis, MO, USA). The supernatants were screened by ELISA. To determine the CDRs of clone #15, V_H_ and V_L_ regions were amplified by PCR with degenerate primers after mRNA extraction and reverse transcription. The amplicons were TA-cloned and sequenced. The CDR analysis was performed by Bio-Peak Co. Ltd. (Takasaki, Japan).

### 4.2. Cell Culture, Expression Vectors and Transfection

HEK293T cells were grown in Dulbecco’s Modified Eagle Medium (DMEM, Glendale, AZ, USA) with 10% fetal bovine serum (FBS; Sigma–Aldrich) and 1% penicillin–streptomycin mixture (Gibco, Waltham, MA, USA).

The protein coding regions of human *CLDN1*, *CLDN4*, *CLDN5*, *CLDN6* and *CLDN9* were cloned into the *Bam*HI/*Not*I site of the CSII-EF-MCS-IRES2-Venus (RIKEN, RDB04384) plasmid.

For transient expression of the target genes (*CLDN1*, *CLDN4*, *CLDN5*, *CLDN6* and *CLDN9*), 5 × 10^6^ HEK293T cells were transfected with 10 µg of the indicated vectors using 30 µg of Polyethylenimine Max (PEI Max, Cosmo Bio, Carlsbad, CA, USA) 8 h after passage. Transfection efficiency was evaluated by Venus expression, with a fluorescent microscope (IX71, Olympus, Shinjuku City, Tokyo, Japan).

### 4.3. Immunoblotting

Total cell lysates were collected with CelLytic MT Cell Lysis Reagent (Sigma) followed by one-dimensional SDS-PAGE and electrophoretically transferred onto a piece of Immobilon (Millipore). The membrane was saturated with PBS containing 4% skimmed milk and treated with primary antibodies. For screening of anti-CLDN6 rat mAbs, supernatants of hybridoma were directly used as primary antibodies, whereas anti-CLDN6 rabbit pAb was diluted at 1:2000 in PBS. The signal was detected by chemiluminescence using HRP-conjugated anti-rat IgG (NA935V, GE Health Care, Chicago, IL, USA) or anti-rabbit IgG (NA934V, GE Health Care).

### 4.4. Tissue Collection, Immunostaining and Analysis

Paraffin-embedded tissue sections were obtained from 173 patients with uterine endometrial cancer who underwent hysterectomy, bilateral salpingo-oophorectomy and/or lymphadenectomy between 2003 and 2012 at Fukushima Medical University Hospital (FMUH) and Iwaki City Medical Center (ICMC). Informed consent was obtained from all the patients. The subjects were limited to patients with confirmed 5-year outcomes and who died due to uterine endometrial cancer and metastasis. The clinicopathological characteristics of patients are summarized in Table 1. Detailed information, including postoperative pathology diagnosis reports, age, stage (FIGO 2008), histological type, histological grade, LVSI, lymph node metastasis, distant metastasis, overall survival (OS) and recurrence-free survival (RFS), was also obtained. The staging of patients between 2003 and 2007 was modified in accordance with the FIGO 2008 system. Distant metastasis was judged by diagnostic imaging. The study was approved by the Ethics Committee of FMUH and ICMC (approval code: 2536/2019-311).

For immunostaining, uterine endometrial cancer tissues were obtained, and the 10% formalin-fixed and paraffin-embedded tissue blocks were sliced into 5-μm-thick sections, then deparaffinized with xylene and rehydrated using a graduated series of ethanol. The sections were immersed in 0.3% hydrogen peroxide in methanol for 20 min at room temperature to block endogenous peroxidase activity. Antigen retrieval was performed by incubating the sections in boiling citric acid buffer (pH 6.0) in a microwave. After blocking with 5% skimmed milk at room temperature for 30 min, the sections were incubated overnight at 4 °C with the primary antibodies. The Histofine SAB-PO kit for rabbit (Nichirei) or VECTASTAIN Elite ABC HRP Kit for rat (Vector Laboratories) was used for 3′,3′-diaminobenzidine (DAB) staining.

Immunostaining results were interpreted by three independent pathologists and one gynecologist using a semi-quantitative scoring system, IRS) [35]. The immunostaining reactions were evaluated according to signal intensity (SI: 0, no stain; 1, weak; 2, moderate; 3, strong) and percentage of positive cells (PP: 0, <1%; 1, 1–10%; 2, 11–30%; 3, 31–50%; and 4, >50%). The SI and PP were then multiplied to generate the IRS for each case. To determine the optical cut-off values of IRS for CLDN6 expression, the receiver operating characteristic (ROC) curve was plotted and analyzed (Appendix A). Based on this analysis, we divided the samples into two groups based on the results of the immunostaining in the tissues: low expression (IRS < 8) and high expression (IRS ≥ 8; Table 2).

### 4.5. Statistical Analysis

We used the chi-squared test to evaluate the relationship between CLDN6 expression and various clinicopathological parameters (age, stage, histological type, histological grade, LVSI, lymph node metastasis, distant metastasis, 5-year OS and 5-year RFS). Survival analysis was performed using the Kaplan–Meier method, and differences between the groups were analyzed using the log-rank test. The Cox regression multivariate model was used to detect the independent predictors of survival. Two-tailed *p*-values < 0.05 were considered to indicate a statistically significant result. All statistical analyses were performed using SPSS version 23.0 software (IBM).

## 5. Conclusions

High CLDN6 expression in endometrial cancer is a biomarker to predict poor prognosis.

## Figures and Tables

**Figure 1 cancers-12-02748-f001:**
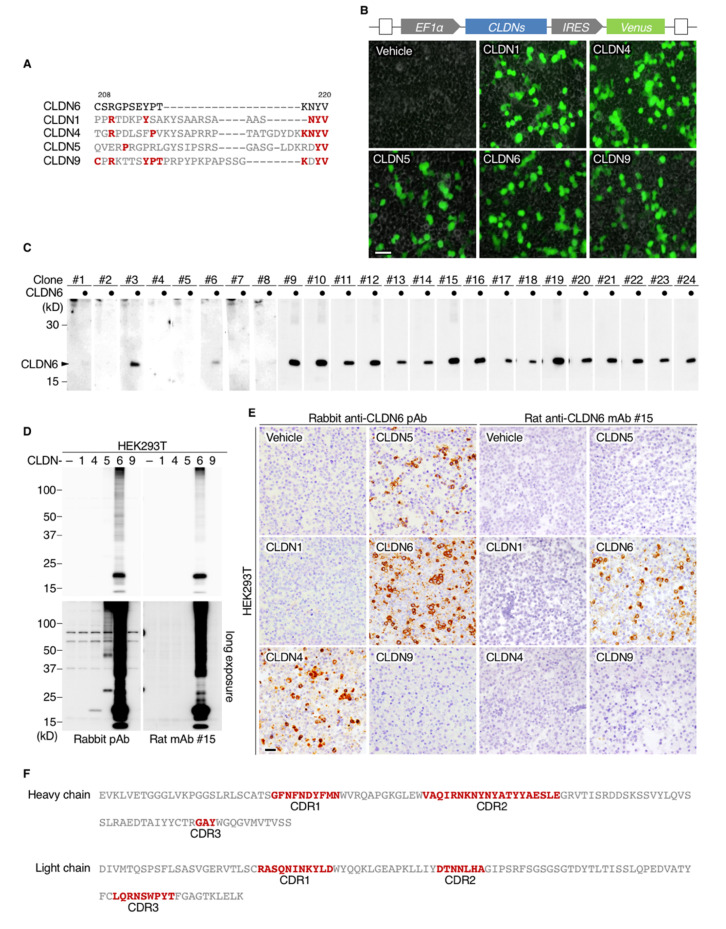
Generation of rat anti-human claudin-6 (CLDN6) monoclonal antibodies (mAbs). (**A**) Amino acid sequences of the antigenic peptide of the C-terminal cytoplasmic domains of human CLDN6 and the corresponding regions of the closely related CLDNs. Conserved amino acids are shown in red. (**B**) The constructs of CLDN1/4/5/6/9 expression vectors and the representative fluorescence images of the transfected HEK293T cells. EF-1a, elongation factor-1a; IRES, internal ribosome entry site. (**C**) Twenty-four hybridoma clones were screened by Western blotting for CLDN6 in HEK293T cells that were transiently transfected with the CLDN6 or empty expression vector. (**D**,**E**) HEK293T cells were transfected with individual CLDN expression vectors, and subjected to Western blotting and immunohistochemical analyses using the indicated anti-CLDN6 Abs. (**F**) The complementarity-determining regions (CDRs) of an anti-human CLDN6 mAb (clone #15). Scale bars, 100 µm.

**Figure 2 cancers-12-02748-f002:**
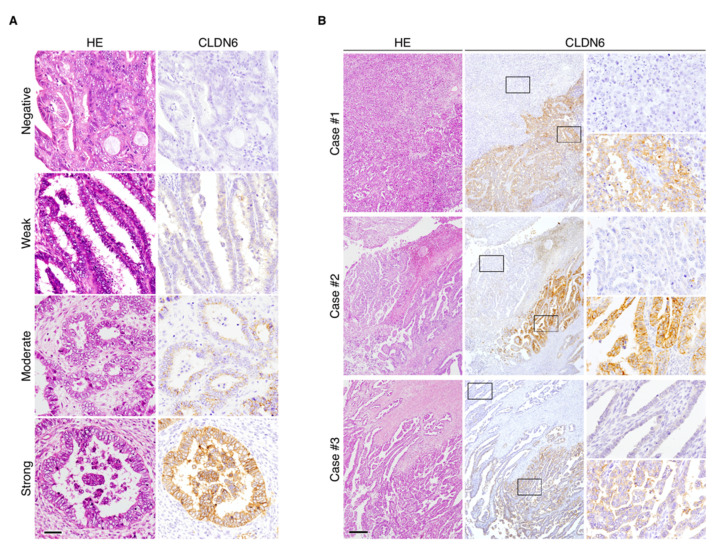
Immunohistochemical staining of CLDN6 in endometrial cancer tissues. (**A**) Representative immunohistological images showing negative/weak/moderate/strong SI for CLDN6 expression in endometrial cancer tissues. HE, hematoxylin–eosin. Scale bar, 50 mm. (**B**) Intratumoral heterogeneity of CLDN6 protein in the subjects with endometrial cancer with high CLDN6 expression. The rectangles indicate CLDN6-positive and -negative subpopulations. Scale bar, 200 µm.

**Figure 3 cancers-12-02748-f003:**
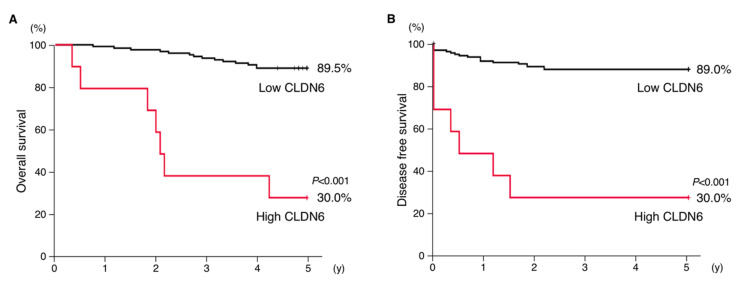
Overexpression of CLDN6 is associated with poor outcome in endometrial cancer patients. The overall (**A**) and 5-year recurrence-free (**B**) survival for high and low expression of CLDN6 in endometrial cancer subjects are indicated.

**Table 1 cancers-12-02748-t001:** Clinicopathological characteristics of patients with uterine endometrial carcinoma.

All Patients	173
Age (years)	33–83 (59 ± 11)
Stage I	138
II	1
II	24
IV	10
Endometrioid	164
Grade 1	110
Grade 2	30
Grade 3	24
Serous	3
Mucinous	2
Clear	4
Relapse (+)	20
Relapse (−)	145
Non-CR	8

CR, complete response.

**Table 2 cancers-12-02748-t002:** Immunoreactivity score (IRS) for CLDN6 expression.

Score	Signal Intensity (SI)		Percentage of Positive Cells (PP)
0	negative		<1%
1	weak		1–10%
2	moderate		11–30%
3	strong		31–50%
4			>50%
**SI × PP**	**IRS**		
0	Score 0	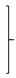	CLDN6 low
1–2	Score 1+
3–6	Score 2+
8–12	Score 3+	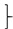	CLDN6 high

**Table 3 cancers-12-02748-t003:** Relation between CLDN6 expression and clinicopathological factors.

Parameter	Total (n = 173)	CLDN6-Low (n = 163)	CLDN6-High (n = 10)	*p*-Value
Age < 50 years	32 (18%)	32 (20%)	0 (0%)	0.122
≥50 years	141 (82%)	131 (80%)	10 (100%)	
Stage I–II	139 (80%)	136 (83%)	3 (30%)	≤0.001
III–IV	34 (20%)	27 (17%)	7 (70%)	
Endometrioid	164 (95%)	156 (96%)	8 (80%)	0.030
Non-endometrioid	9 (5%)	7 (4%)	2 (20%)	
Histological grade 1–2	140 (85%)	136 (87%)	4 (50%)	0.004
3	24 (15%)	20 (13%)	4 (50%)	
LVSI (–)	120 (69%)	117 (72%)	3 (30%)	0.005
LVSI (+)	53 (31%)	46 (28%)	7 (70%)	
N0	145 (85%)	140 (88%)	5 (50%)	0.001
N1	25 (15%)	20 (12%)	5 (50%)	
M0	163 (94%)	156 (96%)	7 (70%)	<0.001
M1	10 (6%)	7 (4%)	3 (30%)	

LVSI, lymphovascular space involvement; N0/1, negative/positive for lymph node metastasis; M0/1, negative/positive for distant metastasis.

**Table 4 cancers-12-02748-t004:** Cox multivariable analysis.

Variable	HR	95% CI	*p*-Value
Age ≥ 50 years	1.61	0.39–6.66	0.513
Stage III or IV	10.93	2.48–48.04	0.002
Histological grade 3	2.18	0.24–3.60	0.091
LVSI (+)	1.91	0.51–7.18	0.340
N1	0.45	0.13–1.61	0.220
M1	4.68	1.57–14.01	0.006
CLDN6-high	3.50	2.42–9.43	0.014

HR, hazard ratio; CI, confidence interval. LVSI, lymphovascular space involvement; N1, positive for lymph node metastasis; M1, positive for distant metastasis.

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
