# Peer review of "Prognostic Significance of Aberrant Claudin-6 Expression in Endometrial Cancer"

_cancers, 2020, doi:10.3390/cancers12102748_

Round 1
Reviewer 1 Report
Thank you for the opportunity to review this manuscript.
The introduction would benefit from a more detailed explanation of the potential role of CLDN6
In the methods- I would be interested to see the ROC curve included to understand how the cut off between moderate and high intensity signally as part of the IHC IRS were determined.
The authors have done well to have a biobank of >150 endometrial cancers, but I feel it is difficult to draw conclusions about the association between CLDN6 and histological type, when the sample is predominantly endometrioid endometrial adenocarcinomas. Additionally, the number of recurrences of deaths are small, so again, the conclusion about CLDN6 as a prognostic marker, particularly with the large number of variables in the multivariable analysis must be considered carefully.
Additionally, what do the authors think about the TCGA molecular classification of endometrial cancer. Rather than histological biomarkers, it seems that prognosis might be better assessed this way? This should at least be included in the discussion.
As this is potentially an upstream marker of the PI3K/AKT/mTOR pathway, whats the benefit of using this biomarker, rather than looking a downstream markers like pAKT?
Finally, the authors could perhaps discuss the role of CLDN6 in the PI3K/AKT/mTor pathway in further detail. I am surprised that this reference
"Knockdown of CLDN6 inhibits cell proliferation and migration via PI3K/AKT/mTOR signaling pathway in endometrial carcinoma cell line HEC-1-B
Reviewer 2 Report
This well-written manuscript by Kojima et al entitled “Prognostic Significance of Aberrant Claudin-6 Expression in Endometrial Cancer” describes a very carefully conducted study on CLDN6 expression in endometrial cancer and its usefulness as a prognostic factor. For their study, the authors raise and characterize a highly specific anti-human CLDN6 monoclonal antibody, which they test meticulously for cross-reactivity. The authors then use this antibody to stain endometrial cancer tissue and to establish a correlation between CLDN6 expression and 5-year survival. In conclusion, the antibody has a high potential to be a valuable tool in cancer staging.
Minor points
line 84: please explain abbreviation “CDR” here, as the Materials chapter (where the abbreviation is explained) is at the end of the article (line 200/201)
Table 3: row “Age >=50”, column “total”: should read 141 (82%) rather than 141 (32%)
line 168 - 173: I find this paragraph hard to understand:
“However, we noticed in the present work that it also reacted with highly expressed CLDN4 and CLDN5 less efficiently than CLDN6, reinforcing the importance of validating the selectivity of each anti-CLDN Ab. Taken together with the finding that CLDN6 is barely expressed in normal adult cells as described above, the established anti-human CLDN6 mAbs would provide powerful tools that selectively recognize CLDN6 protein in a range of cancer tissues.”
do you mean:
“However, we noticed in the present work that it also reacted with highly expressed CLDN4 and CLDN5. This cross-reactive binding is less efficient than the binding to CLDN6, yet it reinforces the importance of validating the selectivity of each anti-CLDN Ab. Considering the higher selectivity of the anti-human CLDN6 mAb and the above-mentioned finding that CLDN6 is barely expressed in normal adult cells, the established anti-human CLDN6 mAbs would provide powerful tools that selectively recognize CLDN6 protein in a range of cancer tissues.”
line 176: heterogeneity of the CLDN6 expression (instead of: heterogeneity on the CLDN6 expression)
